# Zika Virus: A New Therapeutic Candidate for Glioblastoma Treatment

**DOI:** 10.3390/ijms222010996

**Published:** 2021-10-12

**Authors:** Maria Giovanna Francipane, Bruno Douradinha, Cinzia Maria Chinnici, Giovanna Russelli, Pier Giulio Conaldi, Gioacchin Iannolo

**Affiliations:** 1Fondazione Ri.MED, 90133 Palermo, Italy; mgfrancipane@fondazionerimed.com (M.G.F.); bdouradinha@fondazionerimed.com (B.D.); cchinnici@fondazionerimed.com (C.M.C.); 2McGowan Institute for Regenerative Medicine, University of Pittsburgh, Pittsburgh, PA 15219, USA; 3Department of Research, Istituto di Ricovero e Cura a Carattere Scientifico—Istituto Mediterraneo per i Trapianti e Terapie ad Alta Specializzazione (IRCCS ISMETT), 90127 Palermo, Italy; grusselli@ismett.edu (G.R.); pgconaldi@ismett.edu (P.G.C.)

**Keywords:** Zika virus, glioblastoma, glioblastoma stem cells, cancer stem cells, miR34c, neural stem cells, nervous system development

## Abstract

Glioblastoma (GBM) is the most aggressive among the neurological tumors. At present, no chemotherapy or radiotherapy regimen is associated with a positive long-term outcome. In the majority of cases, the tumor recurs within 32–36 weeks of initial treatment. The recent discovery that Zika virus (ZIKV) has an oncolytic action against GBM has brought hope for the development of new therapeutic approaches. ZIKV is an arbovirus of the *Flaviviridae* family, and its infection during development has been associated with central nervous system (CNS) malformations, including microcephaly, through the targeting of neural stem/progenitor cells (NSCs/NPCs). This finding has led various groups to evaluate ZIKV’s effects against glioblastoma stem cells (GSCs), supposedly responsible for GBM onset, progression, and therapy resistance. While preliminary data support ZIKV tropism toward GSCs, a more accurate study of ZIKV mechanisms of action is fundamental in order to launch ZIKV-based clinical trials for GBM patients.

## 1. Introduction

Tumors are one of the leading causes of death worldwide. Glioblastoma (GBM), or grade IV glioma, according to the World Health Organization (WHO) classification [1], is the most malignant tumor occurring in the human central nervous system (CNS), with a median overall survival rate in the United States of about 8 months [2]. Current treatments, such as chemotherapy and radiotherapy, are ineffective in the majority of patients, who ultimately will suffer multiple relapses. GBM resistance is attributed mainly to high levels of cellular heterogeneity and tumor plasticity [3]. Multipotent, self-renewing, and apoptosis-resistant cells were identified in GBMs about 18 years ago [4,5,6]. These cells have been characterized by several groups, and were named glioblastoma stem cells (GSCs), owing to some similarities with normal neural stem/progenitor cells (NSCs/NSPs), including an intrinsic resistance to radiotherapy and chemotherapy treatments [7]. Though considerable controversy remains as to which specific cellular mechanisms drive GBM, there is consensus that the GSC population sustains the long-term clonal maintenance of the tumor, and that the divergence of emerging subclones from a common ancestral clone contributes to tumor evolution and recurrence [8]. In addition to GSCs, tumor-associated fibroblasts, macrophages/monocytes, and endothelial cells also drive intra-tumor heterogeneity and contribute to drug resistance [9].

On a quest to find better treatments for GBM, Zika virus (ZIKV) has emerged as a potential anti-tumor therapy. As a member of the *Flaviviridae* family, ZIKV was first isolated from a rhesus monkey in the Zika forest in Uganda, in 1947 [10]. The ZIKV icosahedral capsid contains a positive-sense, single-stranded RNA genome that spans about 10.7 kb [11]. The transmission occurs through the bite of different species of *Aedes* mosquitoes.

In 2016, an outbreak of ZIKV occurred in the Americas, several Pacific islands, and Southeast Asia, and was followed by the WHO’s declaration of a public health emergency. At that time, ZIKV attracted great attention, due principally to its ability to cross the placental barrier, infect the fetus, and cause severe neurodevelopmental disruptions, including microcephaly [12]. Indeed, ZIKV proteins and RNA were detected in the amniotic fluids, placental tissues, and brain tissues of fetuses with microcephaly [13,14]. Several additional studies have shown the remarkable tropism of different ZIKV strains for NSCs/NPCs, astrocytes, oligodendrocyte precursor cells, and microglial cells [15,16]. Following viral entry into cells, ZIKV disrupts growth and development, causing brain abnormalities. The most vulnerable time for ZIKV infection is between the first and second trimester of gestation, during which ZIKV impairs neurogenesis by reducing the NSC/NPS pool [17]. The causal link between ZIKV infection and neurodevelopment disruption is also supported by animal studies. Intraperitoneal administration of ZIKV in pregnant mice led to radial glial cell death in the fetal cerebral cortex area [18]. Similarly, ZIKV inoculation of rhesus monkeys early in gestation caused alterations of microglial cells and the thinning of the cortical plate in the fetus 3 weeks later [19]. Despite enormous efforts, the complex mechanisms underlying ZIKV-induced microcephaly and other congenital anomalies in humans have yet to be fully elucidated.

In adult patients, ZIKV infection is usually asymptomatic, with only a low percentage of patients (<20%) reporting mild fever, rash, and joint pain for about 7 days [20,21]. Nevertheless, in some cases it can lead to neurological complications [22] or other adverse reactions, including Guillain–Barré syndrome (GBS) [23].

A number of studies have confirmed that ZIKV selectively infects the NSCs in the fetus [24]. This selective targeting of NSCs has encouraged us and other scientists to investigate whether ZIKV might also exert an oncolytic action against GSCs [25,26]. The recent discovery that ZIKV utilizes the neural cell adhesion molecule (NCAM1) receptor [27] to enter the cell [28,29] further suggests that the ZIKV neurotropism could be exploited as a promising strategy for the treatment of GBM. Given that adverse effects of ZIKV are rare in healthy adult humans [23,30], and that in vitro and in vivo models have confirmed the efficacy of ZIKV against GBM, and in particular against GSCs [25,26], ZIKV has the potential to become a novel GBM treatment, with the only exception being, at present, for pregnant women. To date, no clinical trials have been initiated to validate this therapy. In this review, we provide insights regarding ZIKV infection and biology which support the clinical development of ZIKV-based approaches to GBM treatment.

## 2. Glioblastoma Therapies

At present, GBM treatment remains a significant challenge. These tumors grow rapidly and are deeply infiltrating, even if they usually do not spread to other organs. They often invade adjacent brain tissues and are typically localized in the supratentorial region (frontal, temporal, parietal, and occipital lobes) of older patients [31]. A cerebellar location of GBM is more common in younger patients [31], and is associated with a worse overall survival rate [32]. The median survival of GBMs is generally 12 to 15 months from the time of diagnosis [1]. As a result, there is an urgent need to improve therapies against GBM. Treatment options include surgery, radiation, and chemotherapy. Unfortunately, complete surgical removal of GBMs is frequently impossible due to their location and infiltrative nature. Residual masses and tumors that cannot be removed by surgery are commonly treated with high dose ionizing radiation (IR). However, radiotherapy tends to be ineffective, as GBMs are relatively radioresistant [33]. As previously anticipated, self-renewing and pluripotent GSCs might account for the observed radioresistance. GSCs activate the DNA damage checkpoint in response to IR, and repair IR-induced DNA damage more effectively than other tumor cells [34]. Brain endothelial cells may also contribute to GBM radioresistance through the formation of a niche that maintains the GSCs [35]. Finally, in response to IR, GSCs can transdifferentiate into endothelial cells, thus further contributing to tumor revascularization and disease progression [36,37].

In 2005, a trial showed a significant survival benefit when IR was used concomitantly with chemotherapy [38]. This protocol, typically known as the Stupp protocol, has become the standard of care for most GBM patients. As a chemotherapeutic drug, the DNA alkylating agent temozolomide (TMZ) is used. This drug acts by methylating DNA adenine and guanine residues (∼90%) to form N^3^-methyladenine and N^7^-methylguanine, respectively, and to a lesser extent (5–10%) O^6^-methylguanine. The addition of TMZ to IR significantly prolonged survival among patients with newly diagnosed GBM, with a median increase in survival of 2.5 months, or a 37% relative reduction in the risk of death [38]. Moreover, at two years, a clinically meaningful increase in the survival rate was found, from 10% with radiotherapy alone to 27% with radiotherapy plus TMZ [38]. Although TMZ has become a cornerstone of GBM treatment, at least 50% of TMZ treated patients do not respond to the therapy [39]. Resistance to TMZ has classically been linked to the expression of O^6^-methylguanine-DNA methyltransferase (MGMT) [39], which repairs damaged guanine nucleotides by transferring the methyl at the O^6^ site of guanine to its cysteine residues. Accordingly, epigenetic silencing of the *MGMT* gene by promoter methylation is associated with loss of MGMT expression and diminished DNA repair activity, which ultimately correlates with a survival benefit and sensitivity to TMZ treatment [40,41]. Other molecular mechanisms have also emerged as responsible for the appearance of drug resistance in GBM patients, including loss of mismatch repair (MMR) proteins [42], mutation of the tumor suppressor gene *TP53* [43], and overexpression of the *MDR1* gene product P-glycoprotein (P-gp) [44]. Tumors with wild-type (wt) p53 and a functional p53 response to DNA damage are most sensitive to TMZ [45]. In addition to p53, the cyclin-dependent kinase inhibitor p21 is also linked to TMZ toxicity, where tumors with wt p53, but lacking a robust increase in p21 protein level, are resistant to TMZ [45]. By contrast, tumors with a dysfunctional p53 cycle and a weak cell cycle response to DNA damage are extremely unresponsive to treatment [45]. Interestingly, the AMP kinase (AMPK) modulates TMZ-induced p53 activation (phosphorylation at Ser-15/up-regulation) and p21 upregulation [46]. AMPK is considered a metabolic hub. During tumor progression and response to radio- and chemotherapy, the microenvironment changes, causing cells to reversibly switch between glycolysis and oxidative phosphorylation, depending on the availability of oxygen [47]. AMPK is induced by low oxygen and glucose deprivation conditions [48]. Metabolic adaptation is more common for GSCs, and is driven by a complex interplay between microenvironmental cues and the aberrant genetic and epigenetic landscape. Using a longitudinal genomic and transcriptomic analysis of 114 GBM patients, Wang et al. showed a highly branched evolutionary pattern [49]. Despite 45% of the mutations being shared between the diagnosis and relapse samples, the dominant clone at diagnosis was generally not a lineal ancestor of the dominant clone at relapse. Instead, these two clones diverged from a common ancestor more than a decade before diagnosis in most patients. Eleven percent of patients (10/93) exhibited replacement of one mutated version of a gene at diagnosis with another, differently mutated version of the same gene at relapse. This phenomenon was caused by a mutational switching, and occurred preferentially in genes known to play a role in GBM. A few genes appeared exclusively mutated and expressed in recurrent tumors, including *LTBP4* (latent transforming growth factor beta binding protein 4), which encodes for a protein that binds to TGF-β. Silencing *LTBP4* in GBM cells led to TGF-β activity suppression and decreased proliferation, highlighting the TGF-β pathway as a potential therapeutic target in GBM.

The high genomic heterogeneity of GBM tumors has greatly encouraged personalized targeted therapies. Special interest has focused on inhibitors that target receptor tyrosine kinases, such as the epidermal growth factor receptor (EGFR), platelet-derived growth factor receptor (PDGFR), and vascular endothelial growth factor receptor (VEGFR), as well as on signal transduction inhibitors targeting the mammalian target of rapamycin (mTOR) and phosphatidylinositol-3 kinase (PI3K) [1]. Moreover, new drug regimens are being developed to target angiogenesis [50]. The standard-of-care backbone of IR/TMZ therapy has been combined with bevacizumab (Avastin) [51], a humanized monoclonal antibody directed against VEGF, a fundamental regulator of normal and abnormal angiogenesis [52]. However, there are still not enough data to determine whether this combined regimen results in increased overall survival [51]. Moreover, the anti-VEGF receptor-specific small molecule inhibitor AG28262 failed to block the transdifferentiation of GBM cells into endothelial cells, and instead led to an increase in these cells [36].

The U.S. Food and Drug Administration (FDA) recently approved alternating electric fields of intermediate frequencies, also known as tumor treating fields (TTFields or TTF), for the treatment of newly diagnosed GBM. This therapy represents a valuable clinical approach for treating GBM. TTF induces the expression of genes involved in the cell cycle, cell death, and the immune response, regardless of *TP53* status [53], suppressing angiogenesis by downregulating pro-angiogenic factors, including VEGF [54], and might also target vascular niche-associated GSCs [55].

Advances in cancer biology have also resulted in immunotherapeutic strategies to treat GBM. Three major approaches can be distinguished: immune checkpoint blockade, vaccination, and adoptive transfer of effector lymphocytes. Among the drugs targeting immune checkpoints, nivolumab and pembrolizumab were used to target programmed cell death protein 1 (PD-1) in GBM, while ipilimumab was used against the cytotoxic T lymphocyte antigen 4 (CTLA4) [56]. Because PD-1 and CTLA4 are complementary and nonredundant, a combination therapy of nivolumab and ipilimumab was also tested [56]. However, few of these trials were successful, due to intrinsic and adaptive resistance in the early stages of treatment and acquired resistance over the period of therapy, mediated by genetic alternations [57]. In parallel, different cellular vaccines based on T cells, dendritic cells, tumor cells, and natural killer cells were tested in clinical trials for GBM treatment. The recent possibility of transfecting autologous immune cells with pre-manufactured mRNAs encoding full-length tumor-associated antigens (TAAs) overexpressed in a patient’s tumors shows great promise in the advancement of individualized GBM immunotherapies [58]. Another form of immunotherapy consists in the use of oncolytic viruses. Genetically modified oncolytic viruses that express immunomodulatory transgenes are yielding beneficial outcomes [59]. These viruses are typically injected directly into the tumor site or systemically administered, causing direct tumor cell lysis and an immunogenic response. Some viral-based therapies have shown promising results to date in a subset of patients who achieved complete response or stable disease, with long-term overall survival ranging between 4 and 14 years [60]. Below, we will discuss the possibility of using ZIKV as an oncolytic virus for GBM treatment.

## 3. Zika Infection and Viral Involvement in the Immune System

The potential use of ZIKV as an oncoviral therapy against GBM [25,26] has recently been proposed, despite the fact that ZIKV has been associated with transverse myelitis, meningoencephalitis, ophthalmological manifestations, and, most importantly, neonatal malformations and GBS [23,30]. Arboviruses are known for the pathologies they can induce in CNS and the peripheral nervous system (PNS). The association of ZIKV with microcephaly was first observed in 2015 in Brazil [61], when an increasing number of newborns with abnormal brain development and reduced head diameter was reported. Other fetus abnormalities have been observed in pregnant women who suffered a ZIKV infection, including ventriculomegaly, cerebellar and vermis agenesis, cerebral calcifications, and anomalous middle cerebral artery flow [61]. In a C57BL/6 mouse model, it was observed that a main cellular target of ZIKV in fetuses was radial glia cells, the primary NPCs responsible for cortex development [18]. In another mouse model (ICR), following a ZIKV infection, embryonic NPCs underwent apoptosis and inhibition of differentiation, resulting in cortical thinning and microcephaly [62]. Human NPCs (from both adults and fetuses) are also avidly infected by ZIKV [63,64]. In both mouse and human neural cells, the upregulation of genes involved in apoptosis and autophagy has been observed. Moreover, infected NPCs, astrocytes, and microglia produce tumor necrosis factor alpha (TNF-α), interleukin-1 beta (IL-1β), and glutamate, which in turn lead to the death of neighboring, uninfected cells [65,66]. Thus, ZIKV infection not only impairs neurogenesis, but also induces activation of caspase-3 and toll-like receptor 3 (TLR3), leading to cell death [62,63,67]. Currently, the mechanisms that lead to microcephaly are not fully understood due to inherent complexity and several other factors that contribute to this congenital condition. Studies in mouse models have shown the downregulation of genes associated with microcephaly, such as *Aspm*, *Casc5*, *Cenpf*, *Mcph1*, *Rbbp8*, *Stil*, and *Tbr2* [18,62]. Since most of these genes have roles in the cell cycle, their downregulation leads to neuronal growth arrest and cell death. In addition, autoantibodies may contribute to microcephaly, since peptides from the ZIKV polyprotein overlap with many human proteins related to both microcephaly and brain calcification [68]. Moreover, fetuses may be affected by ZIKV infection in other ways. In patients in Brazil, intrauterine growth restriction (IUGR) and excessive liquid accumulation in the fetus were observed [69]. Several miscarriages and stillbirths occurred in women who suffered an earlier ZIKV infection [69,70]. In a heterozygous mouse model (*Ifnar1*^+/−^), IUGR and fetal demise were also observed. Antibody blockade of IFNAR1 in C57BL/6 pregnant mice increased ZIKV trans-placental infection and exacerbated IUGR. In particular, the placental microvasculature was seriously compromised, decreasing the blood flow to the fetus and resulting in IUGR, fetal demise, and ischemia [71]. In babies and infants born with microcephaly from mothers who experienced a ZIKV infection during pregnancy, several ocular defects were observed, such as pigment mottling, chorioretinal atrophy, hypoplasia with double-ring signs, pallor, and increased cup-to-disk ratios.

Another symptom observed in ZIKV-infected patients was GBS, a musculoskeletal paralysis caused by an autoimmune reaction that attacks the PNS [23]. This condition is usually observed 4 weeks after an infection or other stimuli, but in ZIKV patients it arises 6–10 days after infection. Paralysis of the upper and lower limbs are common symptoms, and, in severe cases, the breathing muscles can be affected, leading to the need for mechanical ventilation. Several studies in South American cohorts have shown the frequency of symptoms associated with GBS, ranging from 22% up to 95%. Since ZIKV infects mostly CNS cells, especially astrocytes and oligodendrocytes [72], GBS is most likely caused by inflammatory responses following viral infection. The mechanism responsible for ZIKV-mediated GBS is currently unknown. As mentioned above for microcephaly, it has been observed that several penta- and hexapeptides of the ZIKV polyprotein possess a high level of identity with human proteins involved in demyelination and axonal neuropathies [68], strengthening the hypothesis that ZIKV-induced GBS and microcephaly are probably caused by an autoimmune reaction.

The innate antiviral immune response relies mainly on type I interferons, which restricts viral infection and consequent propagation [73]. Infected cells produce IFN-α and IFN-β, which promote the transcription of type I interferons and other interferon regulatory factors (IRF) in neighboring cells, infected or not, and induce an antiviral immune response. On the other hand, placental trophoblasts produce type III interferons, which can also control ZIKV infection. This virus must, therefore, circumvent IFN-λ to access the fetus and subsequently cause the related congenital pathologies [74]. When knockout mice for *IFN-α* and *IFN-β* receptor 1 (*Ifnar1^−/−^*) or for *IRF3*, *IRF5* and *IRF7* (*Ifr3^−/−^ Ifr5^−/−^ Ifr7^−/−^* triple IRF knockout) were infected with ZIKV, they developed neurological symptoms, e.g., paralysis, and eventually succumbed to the infection [75]. In agreement, an elegant review by Serman and Gack revealed that several ZIKV non-structural proteins interfere with signaling pathways that ultimately lead to IFN-α, IFN-β and the production of related factors [73]. Interestingly, it was observed that endogenous microRNA34a (miR34a) inhibits flavivirus replication, ZIKV included, through the repression of Wnt pathway signaling [76]. Since this pathway is also involved in type I interferon-positive regulation, it is expected that ZIKV infection leads to inhibition of the Wnt pathway, thus impairing the type I interferon antiviral response. In fact, this has been already observed for Dengue virus, another flavivirus [76]. These results confirm the role of type I interferons in controlling the antiviral response against ZIKV.

Thus, the use of ZIKV or related viral therapies to treat GBM in humans, especially in pregnant women, must be carefully evaluated, since many side effects and neurological consequences for both the patient and the fetus can occur.

## 4. Zika and NSCs/NPCs

Both NSCs and NPCs are vulnerable to ZIKV infection. NSCs are a group of multipotent cells that are able to self-renew and proliferate without limit to give rise to the vast array of more specialized cells of CNS and PNS. Conversely, NPCs, have a limited proliferative ability and do not exhibit self-renewal capabilities. In recent years, cells with NSC and NPC characteristics have been derived from induced pluripotent stem cells (iPSCs). Though clear evidence is missing regarding the degree of stemness, induced cells are offering invaluable insights into modeling neurological diseases, especially when cultivated in three dimensions (3D), a particular condition which allows cells to recapitulate aspects of tissue heterogeneity.

Tang et al. [63] first reported the efficient ZIKV infection of iPSC-derived human cortical NPCs with respect to differentiated neurons, which were less susceptible to viral infection. The effects of ZIKV infection included a reduction of NPC viability and growth as well as a down-regulation of cell-cycle-associated pathways. Similarly, the alteration of the molecular pathways involved in neurological diseases, cell death, survival, and embryonic development was observed in human iPSC-derived NPCs and neurons infected with the Brazilian strain, ZIKV-BR [77]. Further evidence that ZIKV abrogates neurogenesis was provided by Garcez et al. [24], who reported a reduced viability and growth of human iPSC-derived NSCs cultured as neurospheres and organoids. An additional study also hypothesized that ZIKV induces a programmed neural cell death, thus causing the disruption of crucial events of early embryonic neurodevelopment [78]. Programmed cell death is essential during morphogenesis, since it helps sculpt brain development. How ZIKV infection induces the cascade of events that may cause impairment of programmed cell death is not fully understood.

The notion that ZIKV preferentially targets human cortical NSCs was demonstrated in cortical brain organoids mimicking a first trimester fetal brain. The results from this 3D model of brain development showed that ZIKV infection reduces proliferation and induces cell death in NPCs, leading ultimately to a reduction of the cortex area, resembling microcephaly [24,79].

A link between ZIKV infection and TLR3-mediated host-innate immune responses was also established. It has been shown that TLR3 serves as a negative regulator of NSC/NPC proliferation in the developing brain. In human cerebral organoids derived from human embryonic stem cells, ZIKV depleted neural progenitors through TLR3 activation, leading to a shrinkage in organoid size reminiscent of microcephaly. Pathway analysis of gene expression changes during TLR3 activation highlighted 41 genes also related to neuronal development, suggesting a mechanistic connection to disrupted neurogenesis [67]. The activation of IFN-associated responses by ZIKV was further reported in primary cultures of human fetal NSC/NPCs [80]. Pharmacological inhibition of the overactivated innate immune responses counteracted ZIKV-induced neurogenesis deficit [80], thus indicating that coordinating the host innate immune responses in NSCs/NPCs after ZIKV infection could be a promising therapeutic approach to attenuate ZIKV-associated neuropathology.

An interesting observation from in vitro studies is that several cells could be the target of ZIKV. While studies in human iPSC-derived neural cells have highlighted the widespread infection and apoptosis of NPCs [78], other studies have indicated that more mature cells, including neurons, radial glial cells, and astrocytes can also be the target [81,82]. Interestingly, radial glia and astrocytes were reported to be more susceptible to infection than neurons [16], because ZIKV replicates more efficiently in undifferentiated compared to differentiated cells [83]. These findings were confirmed in iPSC-derived cerebral organoids [84].

In addition to cell target identification, the characterization of ZIKV entry factors is a key step for the understanding of ZIKV tropism and pathogenesis. Several studies have proposed anexelekto (AXL), a member of the TAM family of receptor tyrosine kinases, as a candidate receptor for ZIKV entry, since its blockage reduces ZIKV infection in NPCs and, subsequently, cell death [16,85,86]. Receptor AXL is expressed at high levels in several cell types susceptible to ZIKV infection, including placental cells, astrocytes, microglial cells, oligodendrocytes, and radial glial cells. Interestingly, the higher expression of AXL in radial glial cells and astrocytes compared to neurons might account for their higher susceptibility to infection [16]. Nevertheless, loss of AXL expression in both human NPCs and cerebral organoids does not impact ZIKV infectivity [87], suggesting that ZIKV entry in these cells may rely on more complex interactions than those associated solely with the AXL receptor. In fact, as described below, ZIKV infection also depends on proteins specifically expressed in undifferentiated cells, such as αvβ5 receptor and SOX2 [29].

Thus, despite the fact that our knowledge of the mechanisms underlying ZIKV transmission and pathogenesis has recently advanced, we still lack a full understanding of the impact of ZIKV infection on the mammalian brain. However, as more widely described below, the selective targeting of undifferentiated cells by ZIKV holds promises in the treatment of GBM.

## 5. Zika and Glioblastoma

As discussed earlier, GBM retains a highly undifferentiated phenotype, with GSCs possessing self-renewal and high tumorigenic properties [4,5,6]. These cells were first described in 2003, when they were selected using in vitro growth conditions analogous to those of normal NSCs [4,5,6]. At present, the origin of GSCs remains unclear, although the general consensus is that they originate from transformed tissue-specific stem cells [88]. Similar to NSCs, GSCs are resistant to apoptotic induction [7] by radiotherapy [34], and chemotherapy [89]. GSC resistance to death, coupled with proliferative, invasive, angiogenetic, and immune evasion properties, contributes to GBM progression and recurrence [90,91,92,93,94], despite aggressive treatment regimens.

Oncolytic viruses, including ZIKV, constitute a promising new strategy for GBM treatment [95,96]. ZIKV is an interesting candidate due to its ability to preferentially infect and kill GSCs [25,26]. Its selectivity towards GSCs seems to depend on the SOX2-integrin–αvβ5 axis [29]. Both SOX2 and αvβ5 are highly expressed by GSCs [97] and correlate with poor prognosis [29]. Thus, they might represent a potential target for future antiviral therapies.

The use of ZIKV in the clinic is supported by the observation that although ZIKV has deleterious effects in fetal development, no major pathologies have been observed in the adult brain [98,99,100,101]. The reason for this different outcome is still unclear, although it can be attributed to the quiescence of NSCs in the adult, and their non-involvement in adult tissue homeostasis [102]. NSCs share many characteristics with GSCs [7], however, the potential risk of ZIKV infection of endogenous NSC populations and the related collateral effects is highly unlikely. A crosstalk between NSCs and GSCs exists within the subventricular zone (SVZ), and this can enhance tumor resistance to therapies [103]. Therefore, the potential targeting of NSCs after ZIKV treatment might even be beneficial. In GSC cultures, ZIKV infection induced miR34c expression [26], which in turn led to a reduction of the anti-apoptotic protein Bcl-2 and the Notch antagonist Numb [26,104], both of which are involved in GSC invasiveness and chemoresistance [5]. Promising results were also obtained in in vivo mouse models of GBM, where ZIKV treatment increased survival, and reduced tumor size [25,105] and metastasis [106]. Importantly, a single intracerebroventricular injection of ZIKV was sufficient to induce such effects [106]. In this regard, it is noteworthy that ZIKV persists in the organism for a long period of time with no evidence of clinical signs. ZIKV can be detected in blood samples more than two months after infection [107,108]. Thus, ZIKV persistence can sustain a protracted oncolytic action, lowering the risk of tumor recurrence and the need for repeated viral infusions. Intrathecal ZIKV injection also led to a reduction of tumor size in an immunocompetent dog model which developed spontaneous intracranial tumors. In these animals, neurological symptoms were improved, and survival was extended, with no clinical virus-related side effects [109]. Moreover, ZIKV modified immune profiling in treated animals, inducing a local immunological response in the tumor mass [109]. In accordance, Nair S. et al. showed that ZIKV treatment induces changes in the GBM microenvironment, increasing local recruitment of CD8^+^ T cells and myeloid cells, and thus contributing to tumor clearance and long-term protection from recurrence [110]. Similarly, Crane AT et al. showed enhanced effector/memory CD4^+^ T cell responses in mice after ZIKV subcutaneous injection, suggesting the use of this virus as a potential adjuvant to vaccine-based immunotherapies against GBM [111]. It is worth noting that ZIKV can promote local immunological response thanks to its ability to alter the integrity of the blood/brain barrier and facilitate immune cell recruitment [112].

## 6. Conclusions and Remarks

The first milestone in tumor therapy can be considered the use of radiotherapy and/or chemotherapy. However, the efficacy of such treatments is reduced by the presence, in virtually all cancer types, of cancer stem cells (CSCs), also referred to as tumor initiating cells (TICs) [7], which escape cytotoxic insults, contributing to the spread and recurrence of the tumor. Evidence suggests that GBM cells with such characteristics, e.g., GSCs, are highly resistant to death stimuli, prompting researchers and clinicians to develop novel therapies that can target them to increase the quality of life and life expectancy of affected patients [7]. Oncolytic virotherapy can meet this challenge. Since China’s first approval for cancer treatment [113], the field of oncolytic virotherapy has expanded at an unparalleled pace. Several oncolytic viruses have been investigated, either in their natural form or genetically engineered. These include herpes simplex virus 1 (HSV-1), adenovirus, vaccinia virus, reovirus, parvovirus, New Castle Disease virus and poliovirus [114,115] (clinicaltrials.gov, accessed on July 2021). In 2015, the U.S. FDA approved a genetically modified, HSV-1–based oncolytic immunotherapy for advanced melanoma (talimogenelaherparepvecImlygic^®^; or T-VEC, previously Oncovex GM-CSF) and currently there are 109 clinical trials for oncolytic virotherapies (clinicaltrials.gov, accessed on July 2021). Twenty of these trials are focused on GBM (Table 1). However, none of these trials uses ZIKV as a therapeutic agent, despite the fact that ZIKV might have an advantage when it comes to the lack of preexisting immunity in the general population worldwide. In fact, all of the above-mentioned oncolytic viruses are geographically widespread, and thus, preexisting immunity in humans against them may exist, which could hamper the success of a therapy based on these viruses [115]. By contrast, ZIKV is restricted to tropical and subtropical areas and, since the 2016 outbreak, no other critical epidemiological situations have been reported (https://www.ecdc.europa.eu/en/zika-virus-disease, accessed on July 2021). Thus, ZIKV might be a powerful tool to induce GSC death in combination with conventional treatments (Figure 1). An engineered form of ZIKV, or ZIKV pseudovirus, has recently been developed [116,117], and holds therapeutic promise for its ability to infect target cells without further replicating, thus avoiding the undesired side effects associated with viral replication and propagation. Another possibility is to exploit ZIKV downstream targets, such as miR34c [26], to treat GBM or other cancer types. This particular miRNA could be successfully carried by viral vectors [26] or extracellular vesicles (Iannolo, unpublished) (Figure 1).

In conclusion, we have reviewed traditional and potential new treatment approaches for GBM. While an effective treatment for GBM is currently not available, advances in cellular and molecular biology are accelerating the discovery of novel therapeutic targets and the design of new drugs and therapies. We further propose a new oncolytic viral therapy based on ZIKV, due to the promising results of this flavivirus against GBM in in vitro and experimental models. While we are aware that ZIKV induces several fetal pathologies, we and others believe that this virus possesses anti-tumoral and immunomodulatory properties that can be harnessed as a therapy against aggressive GBM in adult recipients. Future clinical trials using ZIKV for GBM will confirm the efficacy of this virus as a treatment for one of the most aggressive tumors currently known.

## Figures and Tables

**Figure 1 ijms-22-10996-f001:**
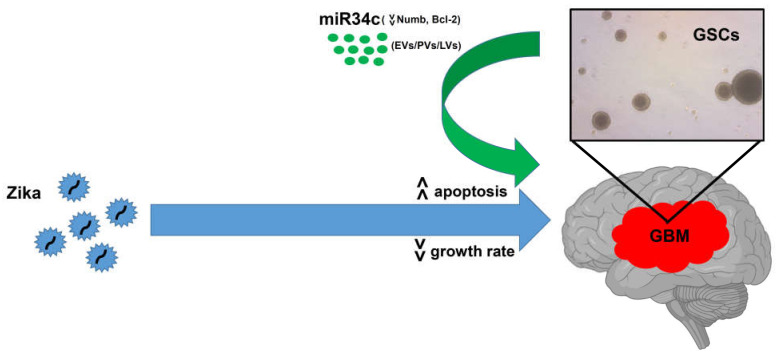
ZIKV-based therapeutic approaches. ZIKV can be directly used against GBM for its specific action against GSCs (blue arrow). ZIKV induces apoptosis and reduces growth rate in GSCs. Additional studies strongly suggest that miR34c is responsible for ZIKV-mediated effects in GSCs, thus supporting alternative therapeutic approaches based on miR34c overexpression by means of extracellular vesicles (EVs), pseudoviruses (PVs) or lentivirus (LVs) (green arrow).

**Table 1 ijms-22-10996-t001:** List of registered clinical trials that use viral agents (alone or in combination with standard therapy) against glioblastoma.

Study Title	Phase	Status	Agent *	Trial ID
A Study of the Treatment of Recurrent Malignant Glioma With rQNestin34.5v.2 (rQNestin)	1	Recruiting	herpes simplex type-1 virus	NCT03152318
Safety and Effectiveness Study of G207, a Tumor-Killing Virus, in Patients With Recurrent Brain Cancer	1–2	Completed	herpes simplex type-1 virus	NCT00028158
Genetically Engineered HSV-1 Phase 1 Study for the Treatment of Recurrent Malignant Glioma (M032-HSV-1)	1	Recruiting	herpes simplex type-1 virus	NCT02062827
Trial of C134 in Patients With Recurrent GBM	1	Recruiting	herpes simplex type-1 virus	NCT03657576
HSV G207 in Children With Recurrent or Refractory Cerebellar Brain Tumors	1	Recruiting	herpes simplex type-1 virus	NCT03911388
HSV G207 Alone or With a Single Radiation Dose in Children With Progressive or Recurrent Supratentorial Brain Tumors	1	Active, not recruiting	herpes simplex type-1 virus	NCT02457845
Oncolytic HSV-1716 in Treating Younger Patients With Refractory or Recurrent High Grade Glioma That Can Be Removed By Surgery	1	Terminated	herpes simplex type-1 virus	NCT02031965
HSV G207 With a Single Radiation Dose in Children With Recurrent High-Grade Glioma	2	Not yet recruiting	herpes simplex type-1 virus	NCT04482933
Virus DNX2401 and Temozolomide in Recurrent Glioblastoma	1	Completed	Adenovirus	NCT01956734
Neural Stem Cell Based Virotherapy of Newly Diagnosed Malignant Glioma	1	Active, not recruiting	Adenovirus	NCT03072134
Oncolytic Adenovirus DNX-2401 in Treating Patients With Recurrent High-Grade Glioma	1	Recruiting	Adenovirus	NCT03896568
Combination Adenovirus + Pembrolizumab to Trigger Immune Virus Effects (CAPTIVE)	2	Completed	Adenovirus	NCT02798406
DNX-2440 Oncolytic Adenovirus for Recurrent Glioblastoma	1	Recruiting	Adenovirus	NCT03714334
DNX-2401 With Interferon Gamma (IFN-γ) for Recurrent Glioblastoma or Gliosarcoma Brain Tumors (TARGET-I)	1	Completed	Adenovirus	NCT02197169
Safety and Efficacy Study of REOLYSIN^®^ in the Treatment of Recurrent Malignant Gliomas	1	Completed	reovirus	NCT00528684
Phase 1b Study PVSRIPO for Recurrent Malignant Glioma in Children	1	Active, not recruiting	Polio/Rhinovirus	NCT03043391
Parvovirus H-1 (ParvOryx) in Patients With Progressive Primary or Recurrent Glioblastoma Multiforme.	1–2	Completed	H-1 parvovirus	NCT01301430
Safety Study of Seneca Valley Virus in Patients With Solid Tumors With Neuroendocrine Features	1	Active, not recruiting	Seneca Valley Virus (SVV-001)	NCT00314925
New Castle Disease Virus (NDV) in Glioblastoma Multiforme (GBM), Sarcoma and Neuroblastoma	1	Withdrawn	New Castle Disease Virus	NCT01174537
Safety Study of Seneca Valley Virus in Patients With Solid Tumors With Neuroendocrine Features	1	Unknown	Seneca Valley Virus	NCT00314925

* the agent used can be genetically modified to increase the oncolytic activity and reduce their toxicity.

## Data Availability

Not applicable.

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
