# Peer review of "Zika Virus: A New Therapeutic Candidate for Glioblastoma Treatment"

_ijms, 2021, doi:10.3390/ijms222010996_

Round 1
Reviewer 1 Report
The paper by Francipane et all describes molecular biology of Zika virus from translational perspective in neuro-oncology. I think it is a well-written paper which deserves to be published. Despite a high octane value of this topic I think authors should convey some message across this paper 1.As stated at the end there are about 20 ongoing clinical trials using oncolytic viruses and it will be great if authors prepare the own-designed table 2.NSC and CSC share several molecular characteristics which prevent Zika to selective infect. That need to be stressed out more clear unless existence of mechanism that prompt Zika virus to distinguish healthy cells from neoplastic cells 3. Topic of ZIka persistence should be specifically highlighted. That opens possibility for Zika persistence in normal cells upon application as vaccineAuthor Response
We thank the reviewer for the suggestions, the text has been modified as requested.

Reviewer 2 Report
The review by Dr Francipane and peers aims to describe the current state-of-the-art about the Zika virus as new therapeutic option for patients with glioblastoma.
The review is well-organized and written, and comprehensive. Authors describe the possibility to use the virus oncotherapy in the treatment of glioblastoma through a deep discussion of the rationale for both malignant gliomas and Zika virus.
The review ha scientific soundness and is supported by appropriate and adequate references.
The topic is interesting and of novelty, due to the possibility to consider Zika virus as potential target for therapy in patients with GBM or recurrences. It may improve the current state-of-the-art in the field of new therapies against the most malignant brain tumours in adults.
Minor points:
- A revision of the English language is advisable. Some misspellings should be amended.
Author Response
We thank the reviewer for the positive comments. The paper has been revised by a language service, and a certificate has been included.

Round 2
Reviewer 1 Report
all comments are being adressed